

# Revisiting historical beech and oak forests in Indiana using a GIS method to recover information from bar charts

Brice Hanberry

Rocky Mountain Research Station, USDA Forest Service, Rapid City, SD, United States of America

## ABSTRACT

Historical GIS involves applying GIS to historical research. Using a unique method, I recovered historical tree survey information stored in bar chart figures of a 1956 publication. I converted PDF files to TIF files, which is a format for a GIS layer. I then employed GIS tools to measure lengths of each bar in the TIF file and used a regression ($R^2 = 97\%$) to convert bar lengths to numerical values of tree composition. I joined this information to a spatial GIS layer of Indiana, USA. To validate results, I compared predictions against an independent dataset and written summaries. I determined that historically (circa 1799 to 1846) in Indiana, oaks were 27% of all trees, beech was 25%, hickories and sugar maple were 7% each, and ash was 4.5%. Beech forests dominated (i.e., >24% of all trees) 44% of 8.9 million ha (i.e., where data were available in Indiana), oak forests dominated 29%, beech and oak forests dominated 4.5%, and oak savannas were in 6% of Indiana, resulting in beech and/or oak dominance in 84% of the state. This method may be valuable to reclaim information available in published figures, when associated raw data are not available.

## INTRODUCTION

Researchers increasingly are applying Geographical Information Systems (GIS) to a range of topics, including historical research (*Gregory & Healey, 2007*). A variety of approaches are available, with more methods and tools continually under development. Currently, there is a great amount of information stored in publications that do not have associated, archived datasets, and in some cases, it may be possible to access that data using GIS.

One example is published information about historical tree surveys. Primarily during the 1800s, the General Land Office divided most of the United States into townships that were subdivided into 36 sections, of 1.6 km (1 mile) squares. Surveyors recorded two to four tree species at section corners and halfway between section corners. These records provide information about forests before sustained Euro-American settlement and disturbance. Despite availability of valuable ecological data that provide a record of historical forests, transferring survey notes from the 1800s to a more accessible format is time-intensive.

In Indiana, although survey notes were transcribed and analyzed by *Potzger, Potzger & McCormick (1956)*, the data currently remain inaccessible, except in form of description,

Corresponding author
Brice Hanberry, bhanberry@fs.fed.us

maps, and graphs. *Potzger, Potzger & McCormick (1956)* presented the approximately 214,500 trees surveyed between 1799 and 1846 in bar chart format by township. To re-transfer data from the historical tree surveys to a GIS layer, with current methods, would take about 1,000 working days at about one township per day, if survey notes are relatively legible. Librarians at Butler University (Potzger collection) and Purdue University (Lindsey papers) were not able to locate any paper copies containing data tables of tree surveys.

*Potzger, Potzger & McCormick (1956)* summarized in bar graph format only the five most common species or genera of American beech (*Fagus grandifolia*), oaks (i.e., primarily white oak, *Quercus alba*, but also including black oak, *Q. velutina*, northern red oak, *Q. rubra*, bur oak, *Q. macrocarpa*, chestnut oak, *Q. prinus*), sugar maple (*Acer saccharum*), upland ash (primarily white ash, *Fraxinus americana*), and hickories (*Carya* spp.). Unlike current forests, many historical forests in the United States were dominated by oaks, pines, or beech, so that information about beech and oaks alone is sufficient to describe forests (*Hanberry & Nowacki, 2016*). Even though exact composition of the approximately 80 tree species (*Potzger, Potzger & McCormick, 1956*) present historically in Indiana remains unknown, species other than oak and beech were minor (2% to 10% of all trees; i.e., hickories, sugar maple, and upland ash) to trace (<2%) components of historical forests at landscape scales. Thus, where beech and oaks were not dominant, they typically were present with many relatively uncommon species (*Blewett & Potzger, 1951*; *Potzger & Potzger, 1950*; *Potzger, Potzger & McCormick, 1956*; *Lindsey, 1961*).

Data in a bar chart format are not useable for other applications, beyond providing a general description, while tabular data are useful, particularly when associated with spatial location. My primary objective was to convert the bar charts presented by *Potzger, Potzger & McCormick (1956)* into tree compositional values joined to a GIS layer of townships. I converted the graphs to a TIF file and then used a regression between lengths in the bar charts and tree composition percentages, using previous work that provided tables of tree composition for townships within six counties (*Blewett & Potzger, 1951* and *Potzger & Potzger, 1950*), to predict composition for the state. I also present ecological information reconstructed from bar charts to validate the method and show the value of recovered data. The study will provide the only source of tree percentages in historical forests of Indiana available currently. This unique method may be helpful for retrieving other valuable datasets collapsed into figures that otherwise would be labor-intensive or impossible to duplicate.

## METHODS

I converted the PDF file of the bar graphs from *Potzger, Potzger & McCormick (1956)* to a TIF file, which is recognized as a raster (ESRI ArcMap software v. 10.3.1, Redlands, CA).

The GIS processing steps are as follows:

1. Convert figure to TIF format
2. Extract by attribute to select dark colored bars (colors up to 190 on 255 color scale)
3. Convert raster to a shapefile

4. Dissolve to turn bar multipart features to singlepart
5. Cut any extraneous shapes that intersect bars
6. Add geometry attributes (extent option) to measure bars
7. Assign species and township (spatial location) information to each bar in attribute table
8. Check data, correct or remove errors
9. Regression between bar lengths and represented values
10. If data are spatial, join to spatial shapefile
11. Validate results with independent or reserved data or written summaries.

I extracted by attribute to select the dark colored bars (colors up to 190 on 255 color scale), converted the raster to a shapefile, and then dissolved to turn multipart features to singlepart (i.e., to make an outline of each bar; Fig. 1). Because the bar graph contained information such as borders and township lines, I cut any extraneous shapes that intersected bars. To determine bar length (along $x$-axis; Fig. 1), I added geometry attributes (extent option).

To provide location, I assigned townships to each row. Because townships presented by *Potzger, Potzger & McCormick (1956)* were simplified, I processed each row to make sure it was assigned to the correct township and range, or removed rows that were indeterminate. For example, townships along the southern river border were difficult to determine correct assignment because rows were offset to avoid placement on the river. I removed townships with no data and small irregular polygons, for example, along rivers, resulting in a final total of 1,000 townships covering 8,875,100 ha (of 9,427,100 ha). I used township and range to join table information to a Public Land Survey System shapefile, which is a spatially correct representation of townships and ranges.

I assigned species information to each row based on row order, provided by *Potzger, Potzger & McCormick (1956)* in the bar graph legends (i.e., oak then hickory, or beech then maple then ash). For about 224 townships, there were <3 rows for the beech, sugar maple, and upland ash. I matched the correct species/genus for each row based on location relative to other rows. Because species/genus was difficult to match in isolated rows, I removed the few small beech, sugar maple, or upland ash rows that represented isolated groves from eight prairie counties of Lake, Newton, Benton, Warren, Jasper, Starke, Pulaski, and White in the Northwest (*Finley & Potzger, 1952*).

I used 40 townships, which were completely within county borders, with known beech, sugar maple, upland ash, oaks, and hickories percentage composition from *Blewett & Potzger (1951)* and *Potzger & Potzger (1950)* for a regression (SAS, version 9.4, Cary, North Carolina; Proc Reg). I regressed percentage against row length to calculate a relationship between percentage and row length ($R^2 = 97\%$). I then used the relationship to predict percent composition for each unknown row length.

For validation, I compared results using root mean square error and mean absolute error to an independent dataset of four complete townships from a GIS layer of GLO tree surveys for the Hoosier National Forest in central southern Indiana (available from the Hoosier National Forest). To compare to written descriptions and maps (presettlement vegetation map by *Lindsey, Crankshaw & Qadir (1965)*; Generalized Presettlement Vegetation Types

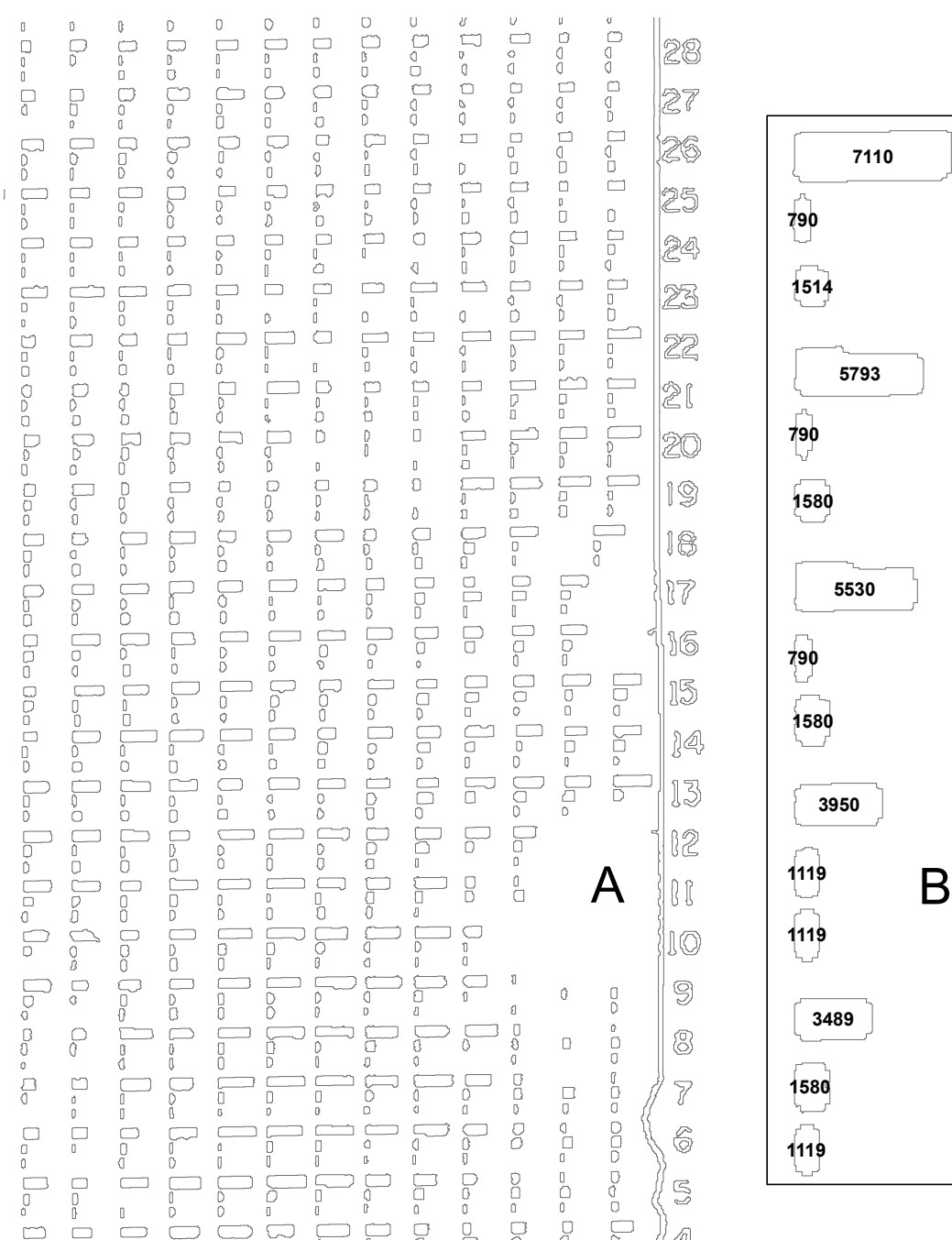

**Figure 1** **Magnified image of original bar graph from** *Potzger, Potzger & McCormick (1956)* **as a GIS layer (A) and example of row length measurements (B).**

of Indiana, Circa 1820, Indiana Department of Natural Resources; http://maps.indiana. edu/previewMaps/Environment/Land_Cover_Presettlement_IDNR.html), I distinguished prairie and oak savannas based on *Potzger, Potzger & McCormick (1956)* and classified forest type by identifying any species or genera that was greater than 24%. If no species

or genera was greater than 24%, then I distinguished the forest type as eastern broadleaf forest, although this may indicate both swamps and upland forests.

I then overlaid beech and oak distributions on 30 year mean precipitation GIS layer for 1981–2010 at 800 m resolution (PRISM; http://www.prism.oregonstate.edu/) and an elevation layer (10 m NED; https://lta.cr.usgs.gov/NED) and derived elevation variables (e.g., slope degrees and terrain ruggedness). I also examined oak and beech distributions over STATSGO soil values (*Miller & White, 1998*; http://www.soilinfo.psu.edu/index.cgi?soil_data&conus&citation) related to water availability: percent clay, available water capacity, permeability, hydrologic soil groups, soil texture class, porosity, and plasticity. I summed values for the soil layers and calculated mean values for each forest type to determine if there was a strong underlying pattern missed by *Lindsey, Crankshaw & Qadir (1965)*.

## RESULTS

To validate regression predictions (regression $R^2 = 97\%$) against another dataset, I compared predictions to four complete townships in a GIS layer of GLO surveys for the Hoosier National Forest. Mean absolute error and root mean square error for predicted values compared to observed values were 2.75 and 3.34 for beech, 1.97 and 2.61 for oaks, 2.49 and 2.64 for hickories, and 3.38 and 3.84 for sugar maple (<1 for ash). Furthermore, tree composition in Hoosier National Forest matched forest classifications.

Maximum predicted composition values by township for oaks (99% of all trees), beech (82%), hickories (42%), sugar maple (42%), and ash (17%) closely matched maximum values listed by *Potzger, Potzger & McCormick (1956)* of 98% composition for oak, 80% for beech, hickories and sugar maple never exceeded 40% composition in any township, and upland ash never exceeded 19% in any township. Overall, for 983 townships excluding 17 prairie townships, oaks were 27% of all trees (area-weighted mean), beech was 25%, hickories and sugar maple were 7% each, and ash was 4.5%. The other approximately 80 species (*Potzger, Potzger & McCormick, 1956*) accounted for 29.5% of all trees.

In this study, beech and/or maple forests dominated (i.e., >24% of all trees) 48% of the state (Fig. 2), while *Potzger, Potzger & McCormick (1956)* identified 39% beech-maple-ash forests and the presettlement vegetation map by *Lindsey, Crankshaw & Qadir (1965)* contained 50% beech forest area. Excluding oak openings, oak forests dominated 31% of the state, while *Potzger, Potzger & McCormick (1956)* identified 28% oak forests and the map by *Lindsey, Crankshaw & Qadir (1965)* contained 30% oak forest area. Oaks were present throughout the state, except in the northwestern prairie portion, and a few isolated townships, resulting in 95 townships of 1,000 townships without oak. Beech was present throughout the state, except in the northwestern prairie portion and southwestern portion, or 179 townships total without beech. The presettlement vegetation map by *Lindsey, Crankshaw & Qadir (1965)* generally matched forest types classified in this study (Fig. 2; http://maps.indiana.edu/previewMaps/Environment/Land_Cover_Presettlement_IDNR.html). *Lindsey, Crankshaw & Qadir (1965)* classified beech-maple and oak-hickory forests based on if one had twice the importance value of the other; otherwise, forests were classified as beech-oak-maple-hickory. The beech-oak-maple-hickory forest type did not

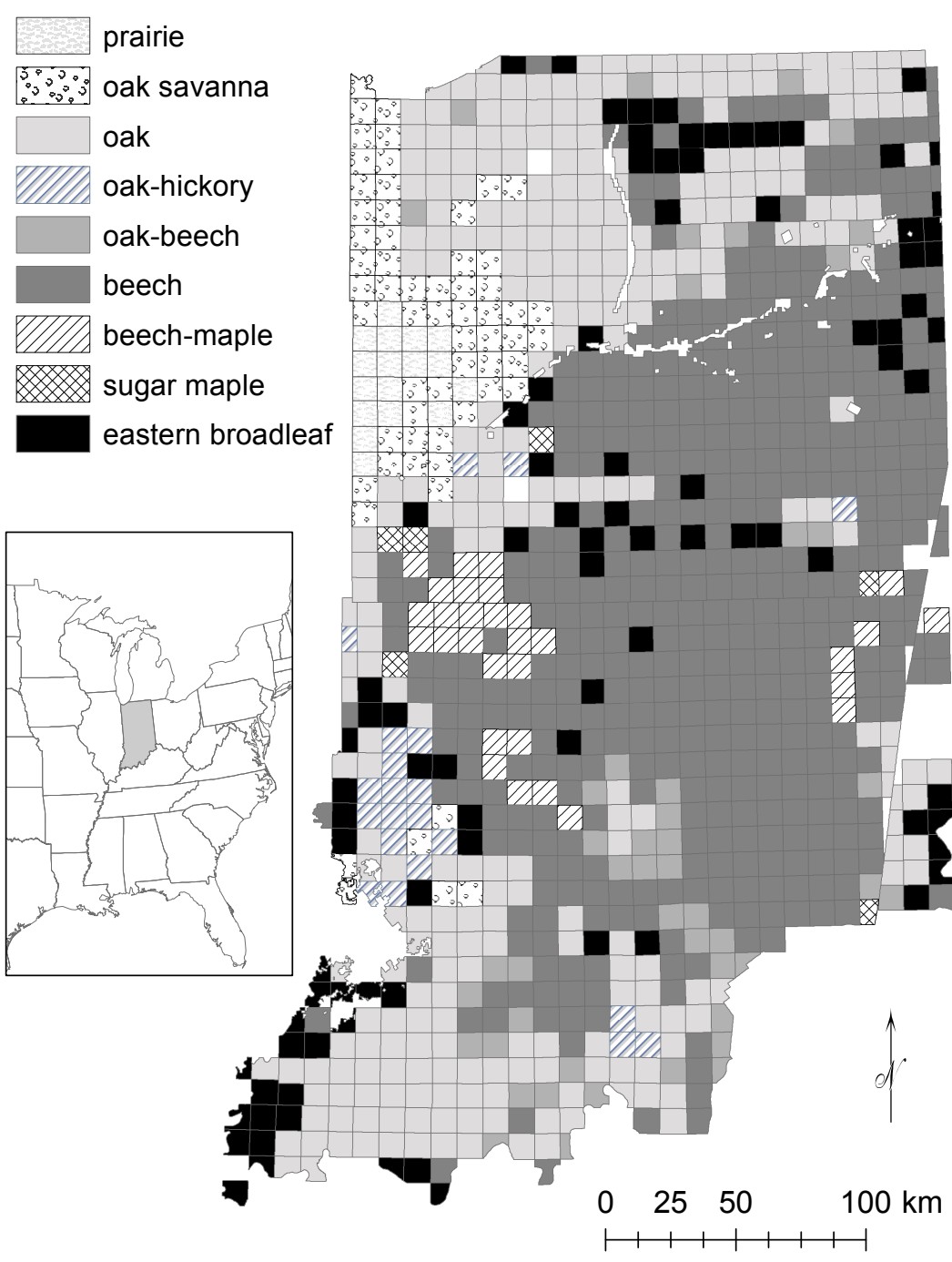

**Figure 2** **Distribution of historical ecosystem types in Indiana.**

match with forests in this study. Also, prairies and wetlands were more extensive than indicated by *Potzger, Potzger & McCormick (1956)*, which were used in this study.

Although the more eastern distribution of beech suggests potential for greater precipitation, 30 year mean precipitation was 107 cm for oak forests and 107.5 cm for

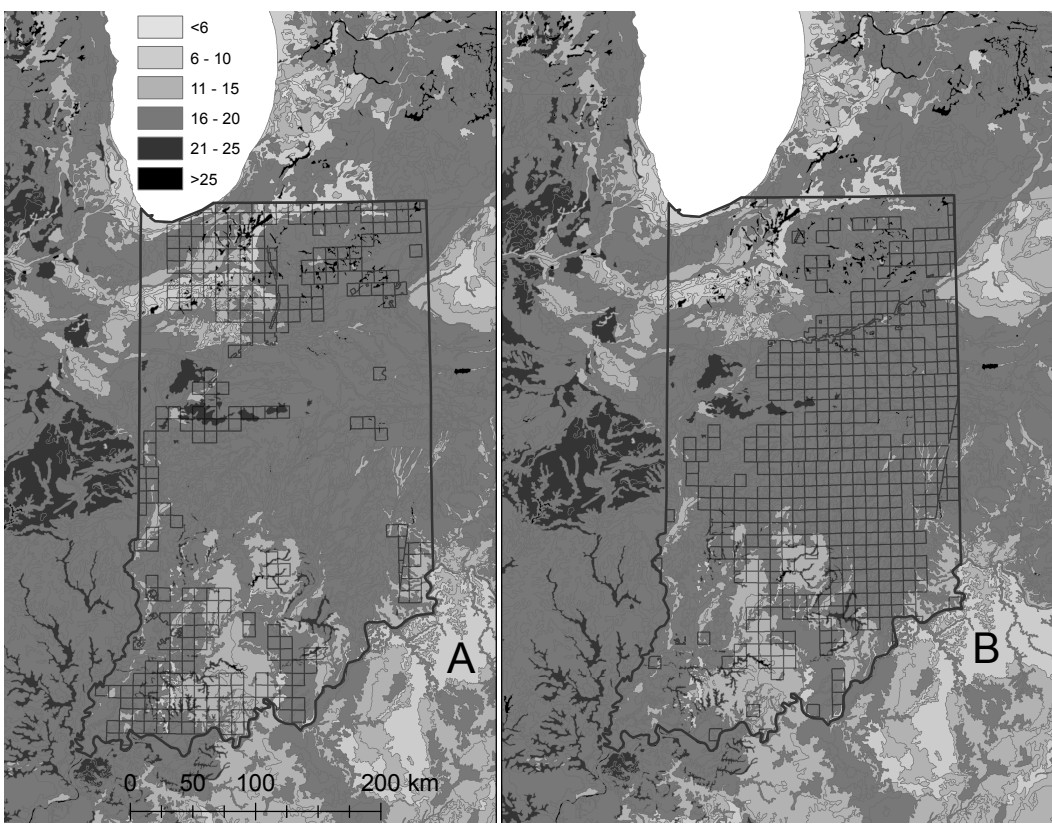

**Figure 3** **Distribution of historical oak (A) and beech (B) forests overlaid on available water capacity.**

beech forests. The lowest value was for prairie at 97 cm, followed by oak openings at 100 cm. Oak-beech, oak-hickory, and beech-maple forests averaged 110–113 cm. Oaks and beech were dominant in both the flat north and central portions of Indiana, as well as the more dissected south. Beech generally was located on moderately wet values for soil variables, but there were exceptions where beech ranged onto other soils, while wetter soil values spread beyond the border of beech distributions to where oaks and prairie also occurred (Fig. 3). Mean value for available water capacity was 16.3 for oak forests and 17.6 for beech forests, but oak-beech forests had the lowest value of all forest types at 15.1. Sugar maple forests had the same value as oak forests, while prairies were about the same as beech forests and oak-hickory forests had greater values (18.4).

## DISCUSSION

It was possible to convert previously published bar charts of historical composition of historically dominant tree species or genera in Indiana (*Potzger, Potzger & McCormick, 1956*) to an accessible GIS layer. Despite the poor quality of the original figure (*Potzger, Potzger & McCormick, 1956*), the TIF image was clear as a GIS layer (Fig. 1). To my knowledge, although this method is relatively simple, there is no documentation of previous use of GIS to recapture data condensed to bar chart format in publications that

do not have archived raw data. Indeed, many current publications do not have associated datasets. The need to reconstruct data from figures also emphasizes the importance of sharing and archiving data, and the developed GIS layer for this study will be archived. This method may be valuable to retrieve a variety of databases published in formats that are not archived or otherwise accessible.

There were numerous sources of agreement between values recovered from bar charts and reported or independent data. There was a strong relationship ($R^2 = 97\%$) between length of rows in bar graphs presented by *Potzger, Potzger & McCormick (1956)* and compositional table values in *Blewett & Potzger (1951)* and *Potzger & Potzger (1950)*. Mean percentage differences between predictions and observed values from an independent dataset transcribed for the Hoosier National Forest were about 2% to 3%. Maximum predicted values by township closely matched maximum values listed by *Potzger, Potzger & McCormick*; (*1956*; 99% compared to 98% for oaks, 82% compared to 80% for beech, 42% compared to 40% for hickories and sugar maple, and 17% compared to 19% for ash). Forest types and areal extent generally agreed with *Potzger, Potzger & McCormick (1956)* and the presettlement vegetation map by *Lindsey, Crankshaw & Qadir (1965)*.

### Recovered ecological information

Use of this method allowed reconstruction of historical composition by species, both spatially in a GIS layer and by state. Indiana is a relatively large state, and inclusion of 8.9 million ha of historical species composition will assist on-going efforts to piece together information about historical forests of the eastern United States (*Hanberry & Nowacki, 2016*). Information about where both oak and beech were dominant particularly may be helpful to understand historical forests.

Excluding prairies, oaks were 27% of all trees (area-weighted mean), beech was 25%, hickories and sugar maple were 7% each, and ash was 4.5%. Beech and/or oak forests (i.e., >24% of all trees) dominated 83% of townships, or 84% of 8.875 million ha due to township area irregularity. Although there were no species/genera >24% of all trees in 92 townships, 36 of these 92 townships contained either beech or oak percentage >20. Sugar maple and hickories, which were the third and fourth most common tree species or genera (*Potzger, Potzger & McCormick, 1956*), were dominant alone in seven townships combined, or 0.7% of all townships. Hickories often are associated with oaks while sugar maple is associated with beech, and yet, historically, hickory and sugar maple abundance was secondary to oak and beech predominance. Indeed, in beech-dominated forests (mean of 44% of all trees), oaks (8%) were as common as sugar maple (8%), with hickories and ash at around 5%. Likewise, in oak-dominated forests (mean of 51% of all trees), beech (6%) was just slightly less abundant than hickories (9%).

There was overlap of beech and oak dominance in 47 townships. In these finer scale locations, oaks and beech may have remained separated spatially by local moisture conditions, at least in topographically variable sites. For example, beech was present on wetter north-facing slopes while oaks were present on drier south-facing slopes (*Potzger, Potzger & McCormick, 1956*); however, *Whitney (1982)* was not able to determine this relationship in northeastern Ohio. Similarly, beech were present in protected valleys or on

lower slopes while oaks were present on drier and more exposed ridges and upper slopes (*Whitney, 1982*; *Cowell & Hayes, 2007*). As for soils, *Whitney (1982)* found that beech were associated with imperfectly drained soils from fine-textured glacial till soils, although he was not able to find congruence between beech forests and Wisconsin drift deposits from glaciation.

At a coarser scale of townships and rather than individual trees, the separation of oak and beech forests may be difficult to explain using precipitation or soil moisture based on topography and associated soils. Precipitation values were approximately 107 cm in historical beech and oak forests, albeit using modern precipitation values for 1981–2010, which are available at 800 m appropriate for resolution within states. Both beech and oak forests were dominant in the flat topography of northern Indiana and the more dissected topography of southern Indiana. Beech forests overall were restricted to moderately wet soils, with exceptions (Fig. 3). Nonetheless, moderately wet soils were abundant in Indiana and the eastern US and extended to where oaks, prairie, and other forest types were present, rather than restricted to beech distributions. *Lindsey, Crankshaw & Qadir (1965)*, after extensive comparison between soils and forest types, were not able to isolate forest types based on soils.

Fire was near-annual in prairie states of the central US and in the southeastern US and became less frequent to the east and north (*Day, 1953*; *Fowler & Konopik, 2007*). *Guyette, Dey & Stambaugh (2003)* quantified a 23 year mean fire interval between 1650 and 1820 near the Ohio River in southern Indiana. Frequent surface fires favored prairies, open forests of savannas and woodlands, and fire-tolerant upland oak species (*Hanberry, Jones-Farrand & Kabrick, 2014*). *Whitney (1982)* noted that fire was recorded in <1% of surveyor records for three counties in Ohio along the border between beech- and oak-dominated forests. However, 67% of fire records occurred in 14% of townships, which were centers of Native American activity. In Indiana, the Wabash Confederacy, for example, probably cleared land through girdling and fire for use along the Wabash River (*Butler, 1895*). *Butler (1895)* wrote about prairies and open forests filled with American bison (*Bison bison*), particularly along river valleys and near West Lafayette (i.e., Ouiatanon) in Indiana, and into Kentucky prairies.

Although oaks in general were the dominant genera (53% of all trees) for the ecological region of the central eastern United States (*Hanberry & Nowacki, 2016*), there also were conditions that favored beech forests interspersed with oak forests. In addition to areas at fine scales that were protected from fire by water, wetlands, moist soils, rugged topography, and rocky outcrops, perhaps fire-protected conditions at landscape scales interacted with reduced or localized Native American fire regimes to produce larger landscapes where fire was less frequent. *Butler (1895)* stated that southeastern Indiana was not settled by American Indian villages, unlike western Indiana along the Wabash River and northeastern Indiana. Central and eastern Indiana thus appeared to be one such stronghold where fire protection allowed beech forests, with some continuity into Ohio, but surrounded to the north, west, and south by oak forests, woodlands, savannas, and prairies.

## CONCLUSIONS

Data published during the 1900s may no longer be effectively available except in figures. I applied GIS to recapture historical data contained in the length of hundreds of column bars in two figures from a 1956 publication. Use of GIS to recover information compressed into figures, with lengths that represent values, is a method that can be applied to other publications where data have been lost.

## ACKNOWLEDGEMENTS

I thank P Hanberry for GIS assistance and anonymous reviewers. This paper may not reflect views of the USDA Forest Service.

### Funding

The author received no funding for this work.

### Competing Interests

The author declares there are no competing interests.

### Author Contributions

- Brice Hanberry conceived and designed the experiments, analyzed the data, contributed reagents/materials/analysis tools, prepared figures and/or tables, authored or reviewed drafts of the paper, approved the final draft.

### Data Availability

The raw data are provided in a Supplemental File.

### Supplemental Information

Supplemental information for this article can be found online at http://dx.doi.org/10.7717/peerj.5158#supplemental-information.

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
