# Peer review of "Revisiting historical beech and oak forests in Indiana using a GIS method to recover information from bar charts"

_PeerJ, doi:10.7717/peerj.5158_

## Round 0.1 · original submission · Major Revisions

I concur with Rev#2 that the paper has a good potential, but it still needs to be restructured: in particular, aims should be more clearly defined and results and discussion properly separated.

What is not yet sufficiently clear to me is if the paper is a methodological one or if it reports a case-study. Both cases have pros and cons and could be valuable, but the aim should be stated more clearly in this sense.

Reviewer 1 ·

Basic reporting

The manuscript is well written. This article meets the basic reporting criteria. The manuscript use ArcGIS to recover information content of historical ecological data. This method should be inspired for who have historical descriptive data. Increasing the possibility of carrying out spatial analysis of historical forest data.

Experimental design

Experimental design is fine. I have only a suggestion for the author. Author had used the bar graphs from Potzger et al. (1956) as the metadata. I think author should provide the supplemental files to show how the PDF file of the bar graphs is. Even use a sample to describe the bar graphs would be useful to future readers. Based on my understanding, many readers have difficulty to access these historical data. In such way, readers would be better understand this method.

Validity of the findings

All data is robust, and analyses are sound. Conclusions and recommendations are well stated. This paper translates historical data into spatial data by GIS technology, which can better analyze the spatial distribution and temporal changes of the forest.

Additional comments

One of my comments is that the authors should provide a flowchart for the methods. A flowchart would help clarify what technique and criterion should being used at each of the step of analysis. In this way, others can better understand and duplicate this method.

Reviewer 2 ·

Basic reporting

see general comments

Experimental design

see general comments

Validity of the findings

see general comments

Additional comments

This paper contains much useful material, but it needs to have a clearer focus before it can be published. At present it is impossible to decide what the author really wants to do. Until the end of the Introduction (or in fact middle of Methods), the paper appears to be focusing on method: how to convert pictorial representations of data processed earlier into useful data similar to what the bar charts had been based on. The Results chapter then describes something different: the pre-European settlement vegetation of the region. Results concerning what the paper claims to be its aim (“my objective was to convert the bar chart ...into compositional values joined to a GIS layer of townships”) show up in the Discussion. In general, the Discussion appears to be a long list of results (mostly answering questions that were not asked by the author) rather than a discussion.
That said, there is a lot to be said for the paper as well. It contains huge amounts of material carefully described (if not much analysed). My advice would be to rethink and restructure the paper. It seems that the actual aim of the paper is to describe the past vegetation of Indiana both for itself and in relation to certain environmental factors. To this aim, turning bar charts into percentages is a mere method. In such a paper, aims should be more clearly defined and justified, and results and discussion properly separated. Alternatively, the paper could retain its methodological focus as well (I personally found this bit quite fascinating), but again aims and structures should be clear. In addition, I think the description of methods is in any case rather difficult to follow with too many things unsaid that are known to GLO specialists but certainly not to the general reader. A figure of the original charts would have been most welcome and so has some form of discussion on the potential amount of similar chart data to contextualize the potential value of this new method.
Lastly, all maps included need a scale and at least one an orientation map showing the position of the study area in north America.

Reviewer 3 ·

Basic reporting

'no comment'

Experimental design

1. Methodology
The author should explain better, perhaps showing some framework of used source, how graphs were transformed into numerical values. Images in the contribution version available online (http://www.jstor.org/stable/pdf/41822538.pdf?seq=1#page_scan_tab_contents) are in very poor condition, and if the same source was used, degree of transformation (R2=97%) may be insufficient.


2. Questions
a) Would it be possible to cross the Vettorializati data of this research with the geological ones pertaining to soil (USGS.gov)? This could probably allow to determine further correlations between soil type and vegetation.
b) In the representation of two tree species within the same square section (e.g. a species with 25% and another with 70%) how can the author recognize/mark this difference?

Validity of the findings

'no comment'

Additional comments

'no comment'

---

## Round 0.2 · accepted · Accept

The reviewer and I agree that you fulfilled the requested changes.

# Reviewer 3 ·

Basic reporting

'no comment'

Experimental design

'no comment'

Validity of the findings

'no comment'

Additional comments

I think this article is now very interesting for PeerJ readers.